# ADAPTIVE SOFTMAX TREES
# FOR MANY-CLASS CLASSIFICATION

## ABSTRACT

NLP tasks such as language models or document classification involve classification problems with thousands of classes. In these situations, it is difficult to get high predictive accuracy and the resulting model can be huge in number of parameters and inference time. A recent, successful approach is the softmax tree (ST): a decision tree having sparse hyperplane splits at the decision nodes (which make hard, not soft, decisions) and small softmax classifiers at the leaves. Inference here is very fast because only a small subset of class probabilities need to be computed, yet the model is quite accurate. However, a significant drawback is that it assumes a complete tree, whose size grows exponentially with depth. We propose a new algorithm to train a ST of arbitrary structure. The tree structure itself is learned optimally by interleaving steps that grow the structure with steps that optimize the parameters of the current structure. This makes it possible to learn STs that can grow much deeper but in an irregular way, adapting to the data distribution. The resulting STs improve considerably the predictive accuracy while reducing the model size and inference time even further, as demonstrated in datasets with thousands of classes. In addition, they are interpretable to some extent.

## 1 INTRODUCTION

Classification problems involving thousands to millions of classes occur naturally in many real-world applications. Examples include predicting the next word in a sentence where the vocabulary size can be in the order hundreds of thousands and categorizing products for e-commerce systems where the number of distinct labels can be in the order of millions. Designing fast yet accurate methods for these type of problems remains an active area of research.

A linear softmax model, either standalone or as the last layer in a neural network, is widely used for general classification problems. Its inference time, however, scales linearly with the number of classes $K$, which makes it very slow for large-$K$ classification problems. A natural way to speed it up would be through conditional computation during inference, but, by design, linear classifiers need to evaluate the score for every class $k$ *no matter the input* $\mathbf{x}$. Decision trees, on the other hand, follow a *single, instance-dependent* root-leaf path during prediction, and their inference time can potentially scale logarithmically with the number of classes $K$. However, traditional axis-aligned trees with constant-label leaves do not produce accurate results for problems with a large number of classes (Choromanska and Langford, 2015).

Recently, Zharmagambetov et al. (2021) propose a novel Softmax Tree (ST) model which strikes a good balance between linear methods and decision trees: the model takes the form of a (hard) decision tree with sparse oblique (linear) decision nodes and small softmaxes at the leaves. To learn these more complex forms of trees the authors adapt a recent Tree Alternating Optimization (TAO) algorithm (Carreira-Perpiñán and Tavallali, 2018), which has the ability optimize various types of tree-based models but only of a fixed structure and size. Experimentally, STs demonstrate much faster inference than the linear classifier and other baselines as well as being very accurate for these large classification tasks. However, a significant drawback of the ST training method is that it assumes a complete tree structure, whose size grows exponentially with depth, and this limits their power. In this paper we propose a new algorithm to train STs of arbitrary structure. The tree structure itself is learned optimally by interleaving steps that grow the structure with steps that optimize the parameters of the current structure. This makes it possible to learn STs that can grow much deeper but in an irregular way, adapting to the data distribution. As we show experimentally, the resulting

STs improve considerably the predictive accuracy while reducing the number of parameters and inference time even further.

After reviewing related work (section 2) and discussing the difficulty of searching over tree structures (section 3), we describe the original softmax tree (ST) model and TAO-based optimization (section 4) and our proposed adaptive softmax trees (AST) (section 5). Then (section 6) we experimentally show the superiority of ASTs over STs and other baselines for several multi-class classification problems with a large number of classes and for language modeling.

## 2  RELATED WORK

**Softmax approximation.** While a softmax linear classifier defines a convex problem with the cross-entropy, it has long been recognized that training it with many classes is a huge computational bottleneck, so that one-vs-all can often be the only affordable option, in part due to its inherent parallelism (Deng et al., 2010). Indeed, even the extremely efficient LIBLINEAR (Fan et al., 2008) implements one-vs-all but not the cross-entropy softmax. And, once trained, inference time in a large softmax is also very large—for example, in a language model having a large vocabulary. Hence, much work has been devoted to approximating the softmax classifier. The Hierarchical Softmax (HSM) (Goodman, 2001) addresses this by using a predetermined tree structure with linear decision nodes and fixed leaf labels (corresponding to the words in vocabulary) to speed up the training of language models. Originally developed for a two-level tree, it has been extended to deeper architectures (Morin and Bengio, 2005; Mnih and Hinton, 2009). The structure of the tree can be random, or based on word similarities (Brown, 1992; Le et al., 2011; Mikolov et al., 2013b), or on word frequencies (Mikolov et al., 2013a; Le et al., 2013), or based on speed-optimal dynamic programming (Zweig and Makarychev, 2013), and optimized for GPUs (Grave et al., 2017). Training HSM-based language models is efficient (usually logarithmic in vocabulary size), but it leads to no speedup at inference time: during prediction an input instance is propagated to all the leaves. Apart from HSMs, other methods of softmax approximation are possible, such as singular value decomposition (Shim et al., 2017) and model compression technicuqes such as pruning or quantization (Deng et al., 2020).

**Decision tree based methods.** Decision trees enjoy fast prediction and interpretability, but traditional methods such CART (Breiman et al., 1984) and C5.0 (Quinlan, 1993) have low accuracy for problems with a large number of classes (Choromanska and Langford, 2015). This is due to a suboptimal training (greedy recursive partitioning) and limited modeling ability (axis-aligned splits and constant-label leaves). The trees can be made more complex by allowing for oblique (hyperplane) splits (Breiman et al., 1984) and small linear classifiers at the leaves (Daumé III et al., 2017). However, this leads to a more difficult optimization problem for which various heuristics and gradient approximations have been proposed (Jernite et al., 2017; Daumé III et al., 2017). Zharmagambetov et al. (2021) adopts a recent Tree Alternating Optimization approach to learn these types of models, but it is limited only to trees of fixed depth and structure. Decision trees are usually ensembled to boost the prediction accuracy but traditional implementations are not suitable for problems with a large number of classes. Si et al. (2017) adapts gradient boosting trees to output $\ell_0$-regularized sparse prediction and applies this to many-class problems. Besides tree-based techniques, other methods exist such as sampling (Joshi et al., 2017) and hashing (Medini et al., 2019).

**Conditional computation.** There is growing interest in having neural networks use only a small portion of its computational graph to enable fast prediction. Although several works Shazeer et al. (2017); Hazimeh et al. (2020); Veit and Belongie (2018) have shown promising results in terms of runtime and accuracy tradeoff, the non-differentiability of the conditional computation makes it difficult to apply continuous gradient-based optimization (Hazimeh et al., 2021). One way to achieve this is to train a continuous model, such as a soft tree, and harden its decisions a posteriori, but this leads to degradation in accuracy, as observed in (Zharmagambetov et al., 2021). In our adaptive softmax trees, conditional computation is built in by design during training and inference.

**Growing NNs and neural architecture search.** The idea of growing the NN architecture by adding more neurons during training has a long history (Fahlman and Lebiere, 1990; Gallant, 1993; Fritzke, 1994; Bruske and Sommer, 1995; Evci et al., 2022). A related, recently very active area that aims to learn an optimal NN structure is Neural Architecture Search (reviewed by Elsken et al. (2019); Ren et al. (2021)). A major issue in learning/growing NN architectures is the vast number of choices: layerwise or depthwise growth, how to connect neurons, etc. With trees the search space more directed: either expanding leaves or pruning nodes. Tanno et al. (2019) adaptively grow and train

soft neural trees (where an input instance follows all root-leaf paths with a positive probability) using backpropagation, but their trees are very small (just a few leaves), thus limiting the potential gains in inference speed.

## 3 OPTIMIZING TREES OVER PARAMETERS AND STRUCTURES

Learning a tree-based model has two important difficulties. One is that the space of tree structures is huge: with $n$ nodes (in total), there are $\frac{1}{n+1}\binom{2n}{n}$ ordered trees (Knuth, 1997), which already exceeds one million for $n = 14$. The other is that a tree (making hard decisions) defines a non-differentiable, highly non-convex optimization problem.

The traditional, widely used approach for learning axis-aligned trees is based on *greedy top-down induction* (Breiman et al., 1984; Quinlan, 1993): starting from the root node, recursive splits are fixed (so they optimize a local purity criterion) until the tree is fully grown. This is usually followed by a form of pruning to reduce overfitting. While suboptimal, this two-step process does a local search over tree structures and can produce adequate results with simple axis-aligned constant-leaf trees, but it works poorly with more complex trees, e.g. with oblique or neural nodes.

The *Tree Alternating Optimization (TAO)* algorithm (Carreira-Perpiñán and Tavallali, 2018), reviewed in section 4, works by optimizing the parameters of each node in alternation, for a tree of given structure. It does a much better job at optimizing a complex tree, as it can monotonically decrease a loss function, regularization and tree model of very general form. It also does a restricted form of structure search: an $\ell_1$ penalty sparsifies the node weight vectors, which can make nodes redundant and thus pruned, resulting in a learned structure that is a subtree of the initial tree. But, beyond that, TAO does not search over tree structures, and in particular it cannot learn a bigger tree than the initial one.

The original Softmax Tree (Zharmagambetov et al., 2021), consisting of a tree with oblique (hard) splits and softmax leaves, relied directly on TAO to optimize the cross-entropy. It used as initial tree a complete tree of depth $\Delta$ and $2^\Delta$ softmaxes each having $k$ classes. By tuning these two hyperparameters $\Delta$ and $k$, it achieved good results on large, many-class datasets. But it has a major limitation: the number of nodes grows exponentially with the depth, which is thus computationally limited in memory and time (to $\Delta \approx 14$ in that paper), which in turn forces the softmaxes to use many classes (large $k$). If the tree was deeper, the softmaxes could be smaller, accelerating the inference. Crucially, depending on the data distribution, the tree may need to be quite deep in some parts and shallow in others, i.e., an unbalanced structure. If we could guess the right structure, we could have TAO use that from the beginning, but guessing it is far from simple. Using, say, a structure from a CART tree does not work at all. This calls for searching over structures properly as proposed in our *Adaptive Softmax Trees (ASTs)*, described in section 5. And, as it turns out, we find in our experiments that ASTs achieve higher test accuracy than using a complete ST of the same depth (which is far more costly).

At the heart of improvement of ASTs is the interplay between tree depth $\Delta$ and leaf softmax width $k$. In a complete ST, the inference time is $\mathcal{O}(D(\Delta + k))$ (actually less if the weight vectors and softmaxes are sparse and some tree paths are shallower than $\Delta$), and typically $\Delta \ll k$. This improves greatly over a single softmax, $\mathcal{O}(DK)$, if $\Delta + k \ll K$. In ASTs, an irregular tree structure makes it possible to reduce $k$ further by increasing $\Delta$ selectively for each branch. Besides, in our ASTs we learn the number of classes $k_j$ for each leaf $j$ automatically, so that some leaves specialize on a few select classes while others handle more classes, which affords further speedups. The inference time is then $\mathcal{O}(D(\Delta_j + k_j))$ for each leaf $j$, and usually larger $\Delta_j$ are associated with smaller $k_j$.

## 4 SOFTMAX TREES (STs) AND TREE ALTERNATING OPTIMIZATION (TAO)

We now describe the Softmax Tree (ST) model and the extension of TAO to train them over a fixed tree structure (Zharmagambetov et al., 2021). Let $\{(\mathbf{x}_n, y_n)\}_{n=1}^N \subset \mathbb{R}^D \times \{1, \ldots, K\}$ be our training set of size $N$ of $D$-dimensional input features and $K$ classes. Write the *Softmax Tree* as $\boldsymbol{\tau}(\mathbf{x}; \boldsymbol{\Theta})$, a rooted binary tree with a set of decision (internal) nodes $\mathcal{N}_{\text{dec}}$ and a set of leaf nodes $\mathcal{N}_{\text{leaf}}$. Each decision node $i \in \mathcal{N}_{\text{dec}}$ has a routing function $g_i(\mathbf{x}; \boldsymbol{\theta}_i): \mathbb{R}^D \to \{\texttt{left}_i, \texttt{right}_i\} \subset \{\mathcal{N}_{\text{dec}} \cup \mathcal{N}_{\text{leaf}}\}$ that sends an instance $\mathbf{x}$ to its left or to its right child. We use oblique (linear) decision nodes: "if $\mathbf{w}_i^T \mathbf{x} + w_{i0} \geq 0$ then $g_i(\mathbf{x}) = \texttt{right}_i$, otherwise $g_i(\mathbf{x}) = \texttt{left}_i$" where the learnable parameters are $\boldsymbol{\theta}_i = \{\mathbf{w}_i, w_{i0}\}$. Note how the routing function makes hard decisions, unlike soft

trees, where an instance $\mathbf{x}$ is propagated to both children with a positive probability. Each leaf $j \in \mathcal{N}_{\text{leaf}}$ contains a predictive function $\mathbf{f}_j(\mathbf{x}; \boldsymbol{\theta}_j) \colon \mathbb{R}^D \to \{1, \ldots, K\}$ that produces the actual output of the tree $\boldsymbol{\tau}(\mathbf{x}; \boldsymbol{\Theta})$ for an instance $\mathbf{x}$. In Softmax Trees, $\mathbf{f}_j(\mathbf{x}; \boldsymbol{\theta}_j)$ takes the form of a small softmax linear classifier: $\mathbf{f}_j(\mathbf{x}; \boldsymbol{\theta}_j) = \sigma(\mathbf{W}_j \mathbf{x} + \mathbf{w}_{j0})$ where $\boldsymbol{\theta}_j = \{\mathbf{W}_j \in \mathbb{R}^{k \times D}, \mathbf{w}_{j0} \in \mathbb{R}^k\}$ are the learnable parameters, and $\sigma(\cdot)$ is the softmax function. The leaf predictor function $\mathbf{f}_j(\mathbf{x}; \boldsymbol{\theta}_j)$ can output only $k$ nonzero probabilities, with $k \leq K$, for a set of $k$ classes (this set is learned); for all the other $K - k$ classes $\mathbf{f}_j(\mathbf{x}; \boldsymbol{\theta}_j)$ assigns exactly zero probability. For problems with a large number of classes we want $k \ll K$ to allow for fast inference. The predictive function of the whole Softmax Tree $\boldsymbol{\tau}(\mathbf{x}; \boldsymbol{\Theta})$ then works by routing an instance $\mathbf{x}$ to exactly one leaf through a root-leaf path of (oblique) decision nodes and applying that leaf's small softmax predictor function. Overall, a ST can be seen as a hierarchical collection of local softmax classifiers each operating on a small subset of classes.

Now we describe how the TAO algorithm applies in learning a ST. TAO is a general method for learning a precisely stated objective function and decision tree model. Unlike CART-type methods, conceptually it works similarly to how one would optimize a (say) neural network: by taking an initial tree structure (network architecture) and parameters (network weights) it performs alternating optimization over the nodes (gradient descent in a neural net) to monotonically decrease the objective function. Unlike with neural nets and soft decision trees, gradient-based optimization is not applicable because hard decision trees are non-differentiable functions. Given a Softmax Tree $\boldsymbol{\tau}(\mathbf{x}; \boldsymbol{\Theta})$ of fixed structure (e.g. a complete tree of depth $\Delta$) and initial parameters (e.g. random), the goal of TAO is to optimize the following objective function:

$$E(\boldsymbol{\Theta}) = \sum_{n=1}^{N} L(\mathbf{y}_n, \boldsymbol{\tau}(\mathbf{x}_n; \boldsymbol{\Theta})) + \lambda \sum_{i \in \mathcal{N}_{\text{dec}}} \|\mathbf{w}_i\|_1 + \mu \sum_{j \in \mathcal{N}_{\text{leaf}}} \|\mathbf{W}_j\|_1 \tag{1}$$

where $L(\cdot, \cdot)$ is the cross-entropy loss, $\boldsymbol{\Theta} = \{\mathbf{w}_i, w_{i0}\}_{i \in \mathcal{N}_{\text{dec}}} \cup \{\mathbf{W}_j, \mathbf{w}_{j0}\}_{j \in \mathcal{N}_{\text{leaf}}}$ are the set of all learnable model parameters, and there is an $\ell_1$ penalty over the weight vectors to promote sparsity via hyperparameters $\lambda, \mu \geq 0$. In general, we use the same regularization value for both decision nodes and leaves $\lambda = \mu$, but in some experiments we explore the effect of the leaf sparsity $\mu$.

The TAO algorithm is based on two theorems. First, the **separability condition** states that eq. (1) separates over a set of non-descendant nodes, e.g. all the nodes at a given depth. This is a consequence of a tree making hard decisions. All such non-descendant nodes can be optimized independently and in parallel. Second, the **reduced problem over a node** states that optimizing the top-level problem of eq. (1) over the parameters of a given node $i \in \{\mathcal{N}_{\text{dec}} \cup \mathcal{N}_{\text{leaf}}\}$ reduces to a simpler, well-defined problem involving only the training instances that currently reach that node $i$ (the *reduced set* $\mathcal{R}_i \subset \{1, \ldots, N\}$). The exact form of the reduced problem differs for leaves and for decision nodes:

- For a decision node $i \in \mathcal{N}_{\text{dec}}$, the top-level problem of eq. (1) reduces to a *weighted 0/1 loss binary classification problem*:

$$E_i(\mathbf{w}_i, w_{i0}) = \sum_{n \in \mathcal{R}_i} c_n \overline{L}(\overline{y}_n, g_i(\mathbf{x}_n; \mathbf{w}_i, w_{i0})) + \lambda \|\mathbf{w}_i\|_1 \tag{2}$$

  where $\overline{L}(\cdot, \cdot)$ is the 0/1 loss, $\overline{y}_n \in \{\texttt{left}_i, \texttt{right}_i\}$ is a pseudolabel indicating the "best" child (i.e., the child that gives the lower value of the loss) for the instance $\mathbf{x}_n$, and $c_n \geq 0$ is the loss difference between the "other" child and the "best" child for the instance $\mathbf{x}_n$. This problem over an oblique node is in general NP-hard, but it can be approximated well with a surrogate loss such as the cross-entropy (i.e., solving a logistic regression). We can ensure monotonic decrease of the top-level objective (1) by accepting the update only if it improves (2) (in practice we find this unnecessary).

- For a leaf node $j \in \mathcal{N}_{\text{leaf}}$, the top-level problem of eq. (1) reduces to a form involving the original loss but only over the parameters of the leaf predictor function $\mathbf{f}_j(\cdot)$ and over its reduced set $\mathcal{R}_j$:

$$E_j(\mathbf{W}_j, \mathbf{w}_{j0}) = \sum_{n \in \mathcal{R}_j} L(\mathbf{y}_n, \mathbf{f}_j(\mathbf{x}_n; \mathbf{W}_j, \mathbf{w}_{j0})) + \mu \|\mathbf{W}_j\|_1 \tag{3}$$

  where $L(\cdot, \cdot)$ is the same cross-entropy loss of eq. (1). Exactly solving this problem would require enumerating all $\binom{K}{k}$ class subsets, but we can approximate this well by picking the top

**input** training set $\{\mathbf{x}_n, y_n\}_{n=1}^N$,
Softmax Tree $\boldsymbol{\tau}(\cdot; \boldsymbol{\Theta})$ of depth $\Delta$.
**repeat**
  update reduced sets $\mathcal{R}_i$ for all nodes $i$;
  **for** $d = \Delta$ **downto** $0$ **do**
    **for** $i \in$ nodes at depth $d$
      **if** $i$ is a leaf
        fit a $k_i$-class linear softmax $\mathbf{f}_i(\cdot; \boldsymbol{\theta}_i)$
        on the top-$k_i$ majority class points
        in $\mathcal{R}_i$ to optimize eq. (3)
      **else**
        fit a weighted 0/1 loss binary
        classifier $g(\cdot; \boldsymbol{\theta}_i)$ to optimize eq. (2)
      **end if**
    **end for**
  **end for**
**until** stopping criterion
**return** trained $\boldsymbol{\tau}(\cdot; \boldsymbol{\Theta})$

---

**input** training set $\{\mathbf{x}_n, y_n\}_{n=1}^N$, initial depth $\Delta_0$,
  softmax contraction coefficient $\alpha \in (0, 1)$
  tolerance ratio for node expansion $\rho > 1$.
$k_0 \leftarrow \alpha K$
initialize $\boldsymbol{\tau}(\cdot; \boldsymbol{\Theta})$ of depth $\Delta_0$ and $k_0$-class leaves;
fit $\boldsymbol{\tau}(\cdot; \boldsymbol{\Theta})$ using TAO;
**repeat**
  update reduced sets $\mathcal{R}_j$ for all $j \in \mathcal{N}_{\text{leaf}}$;
  **for** $j \in \mathcal{N}_{\text{leaf}}$
    initialize ST $\hat{\boldsymbol{\tau}}_j(\cdot; \hat{\boldsymbol{\Theta}}_j)$ of depth $\Delta = 1$ or $2$
    and with $(\alpha k_j)$-class softmax leaves;
    fit $\hat{\boldsymbol{\tau}}_j(\cdot; \hat{\boldsymbol{\Theta}}_j)$ using TAO on $\{\mathbf{x}_n, y_n\}_{n \in \mathcal{R}_j}$;
    **if** $\frac{\text{loss}(\hat{\boldsymbol{\tau}}_j(\cdot; \hat{\boldsymbol{\Theta}}_j))}{\text{loss}(\mathbf{f}_j(\cdot; \boldsymbol{\theta}_j))} < \rho$ **then** expand the leaf $j$
  **end for**
  update the tree $\boldsymbol{\tau}(\cdot; \boldsymbol{\Theta})$ and reoptimize with TAO;
**until** no changes to the tree structure
**return** adaptively grown $\boldsymbol{\tau}(\cdot; \boldsymbol{\Theta})$

Figure 1: *Left*: pseudocode of TAO for learning Softmax Trees. *Right*: pseudocode of the proposed adaptive growth method for ASTs; this uses TAO (left part) as a subroutine.

$k$ majority classes in the reduced set $\mathcal{R}_j$ and training a $k$-class softmax classifier $\mathbf{f}_j(\cdot)$ on them. We solve the resulting $\ell_1$-regularized convex problem using SAG (Schmidt et al., 2017).

While these theorems do not prescribe the order in which the nodes should be optimized, Zharmagambetov et al. (2021) follow a reverse breadth-first search order: all the nodes at a given depth are optimized in parallel, starting from the deepest ones until the root. Each optimization sub-problem involves solving either $\ell_1$-regularized logistic regression or $\ell_1$-regularized $k$-class softmax classifier. As an initial tree, a complete tree of a given depth $\Delta$ is used with initial parameters set either randomly or based on the $k$-means clustering assignment of training points to the leaves. The hyperparameters of the model are the depth $\Delta$ of the tree and the number of classes $k$ in the leaf softmaxes. Fig. 1 (left) outlines the pseudocode of TAO for Softmax Trees. By ensuring that each obtained solution of the reduced problem of a decision node improves upon the previous one, TAO has the guarantee of a monotonic decrease of the objective function (1).

Finally, node pruning occurs automatically because the $\ell_1$ penalty can drive a node's weight vector to 0. This makes the node redundant (it sends all instances to the same child) and can be removed at the end. Thus, the final ST is a subset of the initial (complete) ST.

## 5 ADAPTIVE SOFTMAX TREES (ASTs)

In the previous section, TAO was used on a complete tree of depth $\Delta$. Now, we improve this to explore structures. The basic idea is to use two types of steps. One is a *regular* TAO optimization of a ST of fixed structure (not necessarily complete); this guarantees improvement of the objective defined *globally* over this ST. The other is an *expansion step* on a current leaf, which tries to replace it with a shallow ST (having narrower softmaxes at its leaves). This *local move* can improve the loss function but at the cost of additional decision nodes; we actually expand the leaf if overall we improve, else we do not expand it, and try other leaf. We interleave regular and expansion steps until convergence. Let us see this in more detail.

We first train a shallow (e.g. depth $\Delta = 2$) complete Softmax Tree $\boldsymbol{\tau}(\cdot; \boldsymbol{\Theta})$ with relatively large $k_0$-class softmaxes in the leaves. The number of classes $k_0$ is set such that the total number of predictable classes by the model is at least the total number of classes $K$ in the dataset: $k_0 2^\Delta \geq K$. We then attempt to replace each leaf $j \in \mathcal{N}_{\text{leaf}}$ softmax predictor function $\mathbf{f}_j(\cdot; \boldsymbol{\theta}_j)$ by yet another shallow Softmax Tree $\hat{\boldsymbol{\tau}}_j(\cdot; \hat{\boldsymbol{\Theta}}_j)$ of depth $\Delta = 1$ or $2$, whose leaves contain smaller $\hat{k}_j$-class softmaxes, $\hat{k}_j < k_0$. To control by how much these large softmaxes are reduced we use the following simple heuristic: $\hat{k}_j = \alpha k_0$, where $0 < \alpha < 1$ is the *softmax contraction coefficient hyperparameter*. We obtain this small tree $\hat{\boldsymbol{\tau}}_j(\cdot; \hat{\boldsymbol{\Theta}}_j)$ by fitting it using the TAO algorithm on the training instances that reach the leaf $j$, i.e., on the reduced set $\mathcal{R}_j$. This step can be considered as a recursive application of the Softmax Tree method with the goal of replacing large, flat softmaxes with faster softmax "subtrees". But instead of directly substituting the leaf softmax $\mathbf{f}_j(\cdot; \boldsymbol{\theta}_j)$ with

the tree $\hat{\boldsymbol{\tau}}_j(\cdot; \hat{\boldsymbol{\Theta}}_j)$, we first ensure that the accuracy of $\hat{\boldsymbol{\tau}}_j(\cdot; \hat{\boldsymbol{\Theta}}_j)$ is at least as good as the original softmax $\mathbf{f}_j(\cdot; \boldsymbol{\theta}_j)$ or within a reasonable *tolerance ratio hyperparameter* $\rho$. If this is not the case, the leaf predictor function $\mathbf{f}_j(\cdot; \boldsymbol{\theta}_j)$ remains unchanged. Otherwise, the substitution happens, and this results in the structure change of the original tree model $\boldsymbol{\tau}(\cdot; \boldsymbol{\Theta})$ where it is expanded through the leaf $j$ (the *expansion step*). In this way, after attempting to expand all the leaves $j \in \mathcal{N}_{\text{leaf}}$, and assuming some or all them are expanded, we obtain a deeper Softmax Tree $\boldsymbol{\tau}_{\text{exp}}(\cdot; \boldsymbol{\Theta}_{\text{exp}})$ with smaller leaf softmaxes which has comparable or better training accuracy and faster inference. Now, importantly, we retrain the whole model $\boldsymbol{\tau}_{\text{exp}}(\cdot; \boldsymbol{\Theta}_{\text{exp}})$ globally using TAO (the *regular step*), which will further improve the model accuracy. We repeat these local expansion and global optimization steps until the model converges or some predetermined stopping criterion is reached. Note that if a given leaf $j$ could not grow at one expansion step, it can still grow in the next iteration because of the in-between optimization step which can change the parameters of the whole model. Fig. 1 (right) outlines the proposed adaptive learning algorithm.

The described algorithm can be motivated as performing search through the vast space of different tree structures and parameters. Each leaf-wise local expansion step attempts to improve the model architecture, and the subsequent optimization step of the whole current model attempts to refine the model parameters. This process leads to a better structure and parameter values than the one produced by TAO on a random or heuristic complete tree initialization.

### 5.0.1 COMPUTATIONAL COMPLEXITY

**Training** It is difficult to estimate the training time precisely because of the changing tree structure and softmax sizes. A coarse upper bound results from taking the largest structure and softmax size $k_{\text{max}}$ that occur during training. If we assume that fitting softmax classifiers is linearly proportional to the training set size, then *sequential* optimization of all the leaves is upper-bounded by fitting a single $k_{\text{max}}$-class softmax for the whole training set. But after several expansion steps, the softmax sizes are usually much smaller. Regarding the oblique decision nodes, optimizing *sequentially* all them at a given depth is asymptotically equivalent to fitting a single logistic regression on the whole training set. However, from TAO's separability condition, optimizing all the leaves and all the decision nodes at the same depth can be done *in parallel*, which can bring huge speedups.

**Inference** For the original Softmax Tree (assumed complete), the inference time is $\mathcal{O}(D(\Delta + k))$. Compared to a singlem flat softmax on all $K$ classes, the speed-up is dramatic: $\mathcal{O}(\frac{K}{\Delta+k}) \approx \mathcal{O}(\frac{K}{k})$ if $k \approx \Delta + k \ll K$. For our AST, the inference time for a leaf $j$ is $\mathcal{O}(D(\Delta_j + k_j))$. The improvement is that this results in quite smaller values of $k_j$ at the expense of slightly large values of $\Delta_j$ (thin softmaxes in deep leaves).

## 6 EXPERIMENTS

Our experimental results consistently demonstrate the benefit of our proposed adaptive learning method in learning better Softmax Trees in terms of accuracy, inference time and model size for several benchmarks in classification tasks with a large number of classes and in language modeling. After describing our setup, we first show a detailed comparison of the proposed adaptive growth method against the previous fixed tree approach. We then report benchmark results for document classification and language modeling tasks. Finally, we analyze the produced tree structure and attempt to interpret the model by visualizing it. In this section, "AST" refers to our proposed adaptive learning method, and "ST" refers to the previous fixed tree approach.

**Setup** Unless otherwise stated, we use the following fixed values for these hyperparameters: the initial tree depth $\Delta_0 = 2$ and the depth of expanding subtrees $\hat{\Delta} = 1$. For all other hyperparameters (the sparsity of decision nodes and leaves $\lambda = \mu$, tolerance ratio for node expansion $\rho$ and softmax contraction coefficient $\alpha$) we set them in accordance with cross-validation on a holdout set. All other implementation details including hyperparameter tuning are provided in Appendix A.2.

We compare our results with other baselines specifically developed for problems with a large number of classes. These include RecallTree (Daumé III et al., 2017), LOMTree (Choromanska and Langford, 2015), $(\pi, \kappa)$-DS (Joshi et al., 2017) and MACH (Medini et al., 2019). We use available open source implementations of the above methods or cite their results, where applicable. For the linear one-versus-all classifier we use scikit-learn's implementation (Pedregosa et al., 2011). We report a misclassification error on training and test sets, average in-

Table 1: Comparison between AST and ST. We report train and test errors, depth $\Delta$ of the tree, average inference time per test instance, the number of leaves, average softmax sizes in the leaves $\bar{k}$ and average FLOPs per test instance. For ST we specify its leaf softmax size $k$, for AST the softmax contraction coefficient $\alpha$ and tolerance ratio of node expansion $\rho$. Reported models are trained with $\mu = 0.01$ and models trained with $\mu = 0.1$ are marked with $*$.

| | Method | $E_{\text{train}}\%$ | $E_{\text{test}}\%$ | $\Delta$ | inf.(μs) | # leaves | $\bar{k}$ | FLOPs |
|---|---|---|---|---|---|---|---|---|
| Letter | Softmax | 22.3 | 23.2 | - | 53 | - | - | 416 |
| | ST($k=7$) | 0.52 | 8.33 | 7 | 142 | 128 | 5.27 | 197 |
| | ST($k=5$) | 0.36 | 8.75 | 8 | 98 | 256 | 3.53 | 214 |
| | ST(from AST) | 2.94 | 8.84 | 11 | 86 | 373 | 1.77 | 177 |
| | **AST**($\alpha = 0.85, \rho = 1.2$) | 0.3 | 7.03 | 12 | 43 | 153 | 2.13 | 162 |
| | **AST**($\alpha = 0.75, \rho = 1.2$) | 2.05 | **6.35** | 15 | **9** | 384 | 1.01 | **151** |
| ALOI | Softmax | 10.9 | 13.0 | - | 411 | - | - | 128000 |
| | ST$^*$($k=90$) | 2.01 | 12.3 | 7 | 24 | 126 | 64.9 | 1493 |
| | ST($k=75$) | 3.89 | 12.0 | 6 | 29 | 64 | 74.9 | 1871 |
| | ST(from AST) | 2.37 | 12.8 | 8 | 18 | 177 | 38.4 | 1102 |
| | **AST**$^*$($\alpha = 0.75, \rho = 1.01$) | 1.49 | **9.93** | 10 | **15** | 326 | 23.8 | **1016** |
| LSHTC1 | Softmax | 54.3 | 61.4 | - | 10680 | - | - | 423722 |
| | ST($k=70$) | 14.2 | 62.7 | 7 | 65 | 128 | 70 | 12279 |
| | ST($k=50$) | 6.15 | 61.2 | 8 | 55 | 256 | 49.4 | 9218 |
| | ST(from AST) | 9.36 | 68.7 | 9 | 62 | 511 | 49.7 | 9388 |
| | **AST**$^*$($\alpha = 0.9, \rho = 1.2$) | 16.1 | **60.8** | 10 | **40** | 1006 | 11.5 | **3756** |
| WIKI-Small subs. | Softmax | 42.4 | 50.2 | - | 16500 | - | - | 9214 |
| | ST$^*$($k = 4$) | 48.7 | 51.5 | 8 | 36 | 30 | 4.6 | 691 |
| | **AST**$^*$($\alpha = 0.35, \rho = 1.2$) | 46.3 | **49.5** | 11 | **16** | 73 | 4.1 | **586** |
| | ST$^*$($k = 9$) | 44.1 | 48.3 | 8 | 27 | 50 | 8.0 | 918 |
| | **AST**$^*$($\alpha = 0.39, \rho = 1.2$) | 43.6 | **47.5** | 11 | **12** | 34 | 11.7 | **929** |
| | ST($k = 95$) | 19.7 | 44.1 | 8 | 30 | 256 | 5.65 | 3065 |
| | ST(from AST) | 21.1 | 44.0 | 8 | 19 | 65 | 12.5 | 3296 |
| | **AST**($\alpha = 0.69, \rho = 1.2$) | 37.8 | **42.7** | 13 | **13** | 184 | 2.75 | **1437** |

ference time per sample on the test set and tree parameters (tree depth $\Delta$, average leaf softmax sizes $\bar{k}$ and the number of leaves). We time inference of each sample on single CPU and average it over the whole test set.

## 6.1 THE BENEFIT OF ADAPTIVE GROWTH

We first perform a detailed comparison between the models produced by our adaptive growth method and the previous fixed tree approach. We choose 4 datasets with a large number of classes: WIKI-Small subs., ALOI, LSHTC1 and Letter. The details about them can be found in appendix A.1.

For these sets of controlled experiments we keep node and leaf sparsity parameters $\lambda, \mu$ equal for both ASTs and STs. As stated in previous sections, the AST approach expands leaves unevenly, which produces softmaxes with different number of classes $k$. To ensure that comparison between resulting models is fair and comprehensive, we train STs with biggest $k$ from an AST and cross-validated depth $\Delta$. For WIKI-small we provide a pairwise comparison of multiple STs and ASTs of similar $k$ in Table 1. For example, the softmax size of ST$^*$($k = 13$) and a maximum softmax size of AST$^*$($\alpha = 0.39, \rho = 1.2$) are equal. Then, we use the structure of the final tree from the AST to initialize an ST (referred as "ST(from AST)"). We keep $k$ of leaf softmaxes but reinitialize randomly the weights of linear classifiers in decision nodes and leaves.

Table 1 shows that ASTs considerably outperform STs in test error (up to 5% on WIKI-Small). In many cases the performance of ST is improved as we lower the depth but lowering it too much leads to increase in test error. Note that the depth of STs initialized from the corresponding AST differs because of the post-pruning. Importantly, ASTs have much faster inference (up to 15 times on Letter) and lower FLOPs. Fig. 5 in Appendix contains an additional experiment showing the improved accuracy of ASTs over STs as a function of optimization iterations. These sets of experiments confirm that progressive growth of a tree results in a better local optimum and justifies our approach.

## 6.2 TEXT CLASSIFICATION

We compare our method with other baselines (including ST) on document categorization benchmark WIKI–Small consisting of more than 36k classes. Full dataset contains roughly 380k features and 800k training samples. Setting initial depth of AST to small values (2-3) while keeping $\alpha$ relatively

Table 2: Left: text classification results. Right: language modeling results

**Results on the text classification dataset WIKI-Small.** We report the test error, depth $\Delta$ of the tree and the average inference time per test sample in milliseconds. For STs we specify the leaf softmax size $k$, and for ASTs we specify the softmax contraction coefficient ($\alpha$) and the tolerance ratio of node expansion ($\rho$).

| Method | $E_{\text{test}}(\%)$ | $\Delta$ | inf.(ms) | Train time |
|---|---|---|---|---|
| RecallTree | 92.64 | 15 | 0.97 | 53m |
| one-vs-all | 85.71 | 0 | 10.70 | > 7d |
| MACH | 84.80 | – | 252.64 | 1445m |
| ST(k = 200) | 84.70 | 8 | 0.18 | $\approx$ 1000m |
| $(\pi, \kappa)$-DS | 78.50 | – | 10.33 | - |
| ST(k = 150) | 77.26 | 8 | 0.57 | $\approx$ 1000m |
| AST($\alpha = 0.69$ $\rho = 1.0$) | 77.30 | 12 | **0.03** | $\approx$ 2000m |
| AST($\alpha = 0.60$ $\rho = 1.1$) | **76.21** | 12 | 0.04 | $\approx$ 2000m |

**Results on the language modeling dataset PTB.** Along with the test error, depth $\Delta$ of the tree, and the average inference time per sample in microseconds, we report the average perplexity (PPL) over the test set instances for which the model outputs nonzero probability. The percentage of such instances is shown in parenthesis. For all AST models we set $\rho = 1.0$.

| Method | $E_{\text{test}}(\%)$ | $\Delta$ | inf.($\mu$s) | PPL(%nnz) |
|---|---|---|---|---|
| HSM | 91.1 | 18 | 421 | 575 (100%) |
| one-vs-all | 87.5 | 0 | 705 | 220 (100%) |
| ST(k = 50) | 86.5 | 8 | 58 | 17 (44%) |
| ST(k = 100) | 86.5 | 7 | 58 | 27 (51%) |
| ST(k = 400) | 86.4 | 5 | 64 | 71 (67%) |
| AST($\alpha = 0.3$) | 86.4 | 12 | **17** | 10 (37%) |
| AST($\alpha = 0.4$) | **86.1** | 12 | 18 | 13 (44%) |
| AST($\alpha = 0.5$) | 86.2 | 11 | 19 | 24 (51%) |
| AST($\alpha = 0.75$) | 86.3 | 12 | 20 | 7 (33%) |

high (0.55-0.88) generates extremely big softmaxes in the initial tree, subsequently, causing slow training. Two ways to avoid mitigate this problem: 1) initializing with a bigger initial tree (depth of 5-6) 2) initializing with smaller $\alpha$ (0.007-0.02) while keeping $\alpha$ in expanding subtrees high (0.7). As a result, as AST expands it covers more and more classes.

The left part of Table 2 shows that AST performs better on the test set than most of the baselines. Moreover, our approach shows 6 times faster inference than ST (AST contains on average 44 classes in the leaves). Note increasing number of TAO iterations during leaf expansion or global optimization (or both) may lead to much better results at a cost of training time.

### 6.3 LANGUAGE MODELING

Penn Treebank (PTB) is a popular dataset often used for language modeling. We compare the performance of our AST model on this task against Hierarchical Softmax (HSM), STs and linear one-vs-all clasifiers. The details regarding the dataset preprocessing, implementation of the baselines and the hyperparameter tuning can be found in Appendix A.3.

Perplexity score PPL $= \exp(-\frac{1}{N} \sum_{n=1}^{N} \log Pr(y_n|\mathbf{x}_n))$ can be undefined for models that can output exacly zero probability. This can happen with STs where an instance $\mathbf{x}$ reaches a leaf whose softmax does not specialize in the true class $y$, and thus gets $Pr(y|\mathbf{x} = 0)$. Therefore, in estimating the PPL we only include the instances for which the model outputs nonzero probability. Although a linear classifier provides positive probability for all the classes, it could not predict correctly 58% of all $K \approx 6k$ classes on both training and test sets, i.e., the outputted score $Pr(y|\mathbf{x})$, though being positive, was not a maximum, not even in the top-10 for many instances. For our AST models it is possible control the percentage of points for which the model outputs positive probability by tuning the hyperparameter $\alpha$, which appendix A.4 explores in-detail.

The left part of Table 2 shows the results on PTB. It is clear that our method outperforms other baselines in both top-1 test error and in inference time by a considerable margin. The performance of AST can be even further improved by more optimization iterations.

### 6.4 TREE STRUCTURE AND INTERPRETABILITY

Fig. 2 shows how the number of classes present in the leaves changes with depth. Theoretically the number of classes in the leaves should only monotonically decrease with depth. Such deviations due to two reasons: 1) number of classes in the reduced set of the given depth is lower then theoretical upper limit; 2) post pruning brings leafs closer to root.

The intrinsic tree structure of our model allows for its interpretation by visualizing its structure and parameters. To show this, we train an AST on a small subset of Amazon Reviews dataset (He and McAuley, 2016) which contains text reviews for the products in the Amazon website. From four high-level product categories (Sports, Toys, Home, Tools) we select 50 subcategories with the highest number of reviews. We select up to 300 reviews from each subcategory, and extract tf-idf transformed bag-of-words features. This results in a dataset of size about 60k with features of size

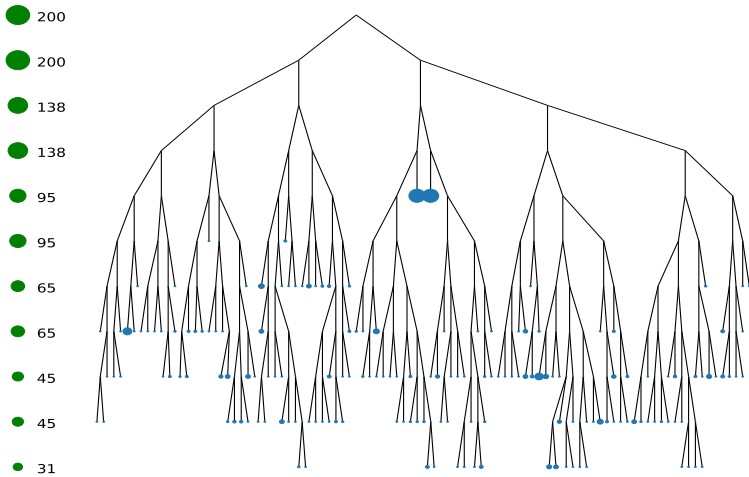

Figure 2: AST for the Wiki-Small subs. dataset. Size of the blue nodes shows the actual number of classes in the leaves after pruning. Green shows theoretical max values at each aligned depth.

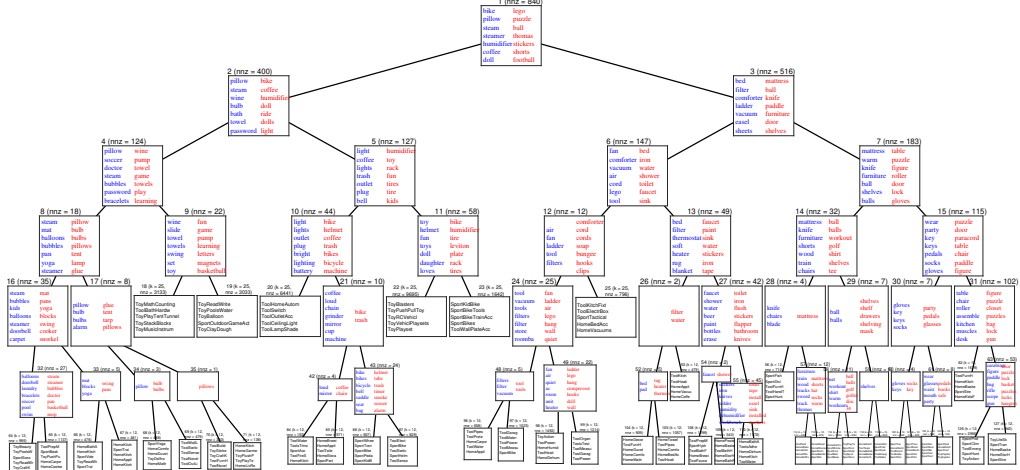

Figure 3: Visualization of an AST tree for a subset of Amazon Reviews dataset.

of 11k and 200 classes. We keep 20% of the dataset as test set, and train a relatively smaller AST on this problem to be able to visualize it in the paper. Initial tree has depth $\Delta_0 = 2$, and $\alpha = 0.25$, and we limit the expansion steps up to 2. The resulting tree has accuracy of 53%, and depth $\Delta = 6$, and is visualized in fig. 3. The general observation reveals the hierarchical structure, where we can observe some subtrees specializing on similar groups of classes (e.g. decision node 9 specializes mostly on Toy classes). Looking at the decision node weights in the root-leaf path one can get a local interpretation of why the tree sends a point to that particular leaf, and a small and sparse softmax model at the leaf is also considered interpretable. Another key observation is that for the most part similar classes tend to be grouped in the same leaf, which is quite remarkable given that the tree is initialized randomly unaware of any class information.

## 7 CONCLUSION

Softmax Trees are an effective model for many-class problems which capitalize on the conditional computation of decision trees and the ability to define local softmax classifiers that handle small subsets of classes, both of which make inference very fast. However, the existing algorithm operates on a fixed, complete tree, which considerably limits the depth of any individual leaf and results in local softmaxes being wider than necessary. Our Adaptive Softmax Tree solves this by learning the tree structure: it interleaves local expansion steps that turn a wide softmax into a Softmax Subtree with thinner softmaxes, with a global TAO-based optimization of the entire tree. Our experimental results convincingly show this results in improved accuracy, inference time and model size.

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

## A    APPENDIX

### A.1    DATASETS

| Dataset | $N_{\text{train}}$ | $N_{\text{test}}$ | $D$ | $K$ |
|---|---|---|---|---|
| Letter | 16 000 | 4 000 | 16 | 26 |
| ALOI | 97 200 | 10 800 | 128 | 1000 |
| LSHTC1 | 80 552 | 19 873 | 271 022 | 2657 |
| WIKI–Small (subs.) | 20 000 | 10 000 | 54 188 | 200 |
| WIKI–Small | 796 617 | 199 155 | 380 078 | 36 504 |
| PTB | 400 097 | 34 633 | 150 | 5 970 |

Table 3: Datasets used in the experiments: number of train and test instances ($N_{\text{train}}$, $N_{\text{test}}$), number of features $D$, number of classes $K$.

To create the subsampled Wiki-Small dataset, we randomly select equal number of samples from each class to avoid imbalance. This is done with two purposes: 1) smaller dataset allows to train for much higher number of iterations (to eliminate undertraining); 2) reduces time of a single experiment which facilitates more precise hyperparameter search. Further, we remove features that remain constant for all the training and test points. As a result, input features of the subsampled WIKI-Small

have $D = 37k$ dimension represented as normalized bag-of-words. For LSHTC1, we eliminate all classes that contain less then 10 samples per class. We used tf-idf feature representations of $D = 271k$ dimension and $K = 2657$ classes. Table 3 summarizes the used dataset statistics.

## A.2 IMPLEMENTATION DETAILS AND HYPERPARAMETERS

Both ST and AST were implemented in Python 3.8.10 and parallelized using Ray 2.2.0 (Moritz et al., 2018). $l_1$-regularized logistic regression in the decision nodes implemented in scikit-learn (Pedregosa et al., 2011) was solved using LIBLINEAR (Fan et al., 2008) and in the leaves - using SAGA (Defazio et al., 2014).

For ST, a search of hyperparameters was performed on separate holdout set. We found that $\lambda = 0.01$ leads to best performance for most datasets and $\lambda = 1$ - for WIKI-Small. For smaller datasets we set the number of TAO iterations high (up to 100), we report average of 5 runs and set number of LIBLINEAR and SAGA iterations to 100. For larger experiments, TAO iterations are set to 40 with average of 3 runs and the number of LIBLINEAR and SAGA iterations is set to 100 and 50 respectively. Trees were initialized using random initialization as well as k-means initialization described in Zharmagambetov et al. (2021).

For AST, both leaf and node sparsity parameters were cross-validated separately on a range between 0.01 and 100. It was found that for Letter and subsampled WIKI-Small $\lambda = \mu = 0.01$, and for ALOI and LSHTC1 $\lambda = \mu = 0.1$ performs best. For large datasets $\lambda = \mu = 1$ produces best results. We initialized initial tree as well as stumps during expansion using median split. This way nodes have almost the same number of samples and training in parallel becomes faster and generally produces better accuracy. The number of LIBLINEAR and SAGA iterations is similar to one in ST. One way of speeding up expansion process is to use weight matrix of expanding decision node to warm-start optimization in leaves. This way SAGA converges much faster for the same tolerance. Number of TAO iterations in during the expansion is set to 10 and to 15 during global reoptimization, but in many cases it converges faster.

## A.3 LANGUAGE MODELING EXPERIMENTS

The Penn Treebank contains around 1M tokens and vocabulary size of 10k words. Similar to Zharmagambetov et al. (2021), we filter out rare words and obtain word embeddings using pre-trained GloVe (Pennington et al., 2014). We predict the next word based on previous 3 words. To form a preprocessed dataset, we simply concatenate word vector representations. As a result preprocessed PTB consists of roughly 400k training samples, 150 features and 5970 classes. For baselines, we used one-vs-all classifier from scikit-learn with $\ell_1$ regularization $\lambda = 1$ and Hierarchical Softmax from Mikolov et al. (2013a) implemented in Pytorch. We further compare AST and ST of different leaf softmax sized ($k$) to show that AST wins not only in terms of top-1 test error but is up to 4 times faster in inference.

## A.4 CONTROLLING LEAF SOFTMAX SIZES FOR LANGUAGE MODELING

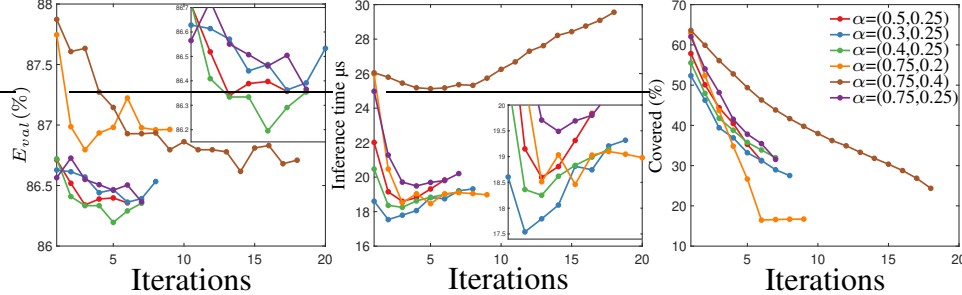

Figure 4: Top-1 error, average inference time and percentage of covered classes for AST of different $\alpha=(\alpha_0, \alpha)$ on PTB dataset.

Fig. 4 shows dependence between the proportion of covered samples of different AST models. We found experimentally that best validation performance is achieved when $\lambda$, $\mu$ and $\rho$ are set to 1. Fig. 4 shows that for high values of $\alpha_0$ and $\alpha$ ($\alpha_0 = 0.75$, $\alpha = 0.4$) tree grows extremely deep (high number of expansion steps) while maintaining relatively big softmax in the leaves. Moreover, the fig. 4 highlights that as softmax size decreases with tree depth so does the inference time, however at some point it starts to go up again. Since time it takes to propagate a sample to the leaf overtakes the time of matrix multiplication in softmax there is an optimal depth of the tree for which inference is the fastest. On the other hand, for very small $\alpha$ ($\alpha_0 = 0.75$, $\alpha = 0.2$) softmax size decreases much faster with tree depth resulting in a small tree with very small number of classes in the leaves. Experimentally we found that such trees do not generalize very well and typically have low class coverage. We can specify number of expansion steps (maximum depth) of the tree to control the minimum coverage and inference time.

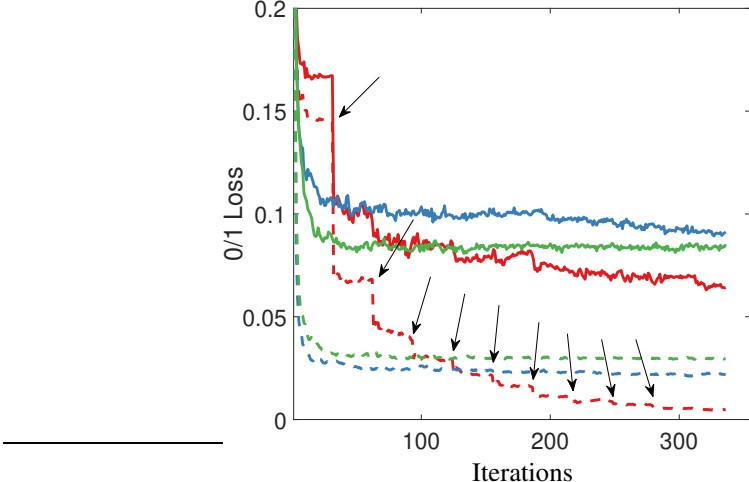

Figure 5: Final model 0-1 Train (dashed line) and Test (solid line) loss comparison against complete Softmax Tree. Arrows show expansion of the tree. Blue line shows performance of ST, Green - ST(AST) and Red - AST. This shows that adaptive growth gradually enhances the performance of the model on both train and test tests (red solid and dashed lines). On the other hand, ST initialized randomly (blue line) or on the final structure of AST (green line) is unable to improve after certain number of iterations.

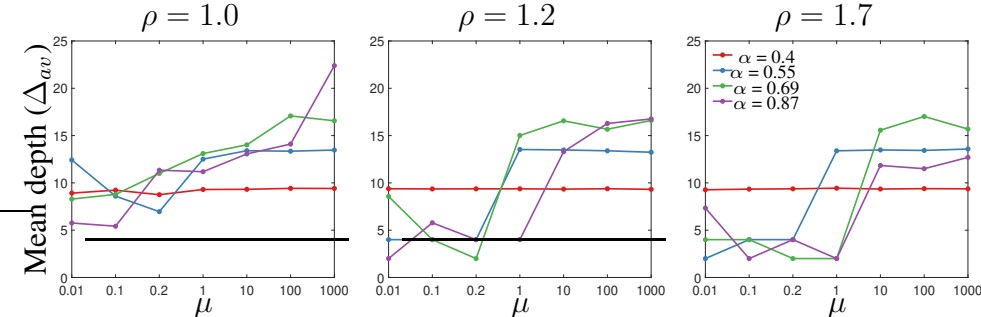

Figure 6: Comparison of average tree depth $\Delta_{av}$ vs softmax regularization parameter $\mu$ for different values of $\rho$ and $\alpha$, $\lambda = 0.01$

We examine effect of softmax contraction coefficient ($\alpha$), tolerance ratio for node expansion ($\rho$) and leaf sparsity parameter ($\mu$) on the final tree structure. We conduct this set of experiments 5 times on subsampled WIKI–Small dataset to eliminate the effect of noise and any inconsistencies.

We measure average tree depth $\Delta_{av}$ over depth of each leaf in final AST. Fig. 6 shows that $\Delta_{av}$ tends to increase as we increase $\mu$. More sparse softmax in the leaves means expanded subtree is more likely to perform better on the reduced set. It subsequently leads to more leaves being expanded

on the current depth. Maximum depth, on the other hand, does not grow significantly. Fluctuations of average depth as we increase leaf sparsity can be explained by good local optimum for given $\mu$. In general it was found that as the number of TAO and SAGA (solver for softmax classifier) iterations increases lines become more smooth.

