# OpenReview forum: "Adaptive Softmax Trees for many-class classification"
_ICLR.cc/2024/Conference — ICLR 2024 Conference Withdrawn Submission_

### Official Review · Reviewer_P37h · 2023-10-29

**Soundness:** 3 good
**Presentation:** 4 excellent
**Contribution:** 2 fair
**Rating:** 6
**Confidence:** 4

**Summary:**

The paper presents Adaptive Softmax Trees, an extension of Softmax trees, designed for many-class classification tasks. Similar to Softmax trees, the method is using Tree Alternating Optimization (TAO) to learn decision trees with sparse hyperplane splits and small Softmax classifiers in the leaves. Different to Softmax trees, the current method does not assume a complete tree. Instead, it grows the tree iteratively and therefore is able to learn deeper trees that are not complete but adapted to the data distribution.

**Strengths:**

Strengths:
- Extension of the recently proposed Softmax trees that is designed for many-class classification tasks (typically in NLP) and is able to reduce the inference time and model size.
- Experiments show the proposed approach leads to significantly shorter inference time and better generalization (lower testing error).
- The paper is clear, written well, and easy to follow.

**Weaknesses:**

Weaknesses:
The main weakness is the somewhat limited technical novelty: this is a relatively straightforward extension of Softmax trees, combining it with iterative expansion/growing of leaves. The key contribution is described in Section 5, and the majority of the technical content is an extensive summary of related previous  works (primarily, Tree Alternating Optimization and Softmax Trees). The rest of the paper is dedicated to different experiments that, as noted in Strengths, shows clear empirical improvement over previous work, however there is no significant technical insight beyond that.

**Questions:**

The paper is clear and I do not have any questions

---

### Official Review · Reviewer_qCCg · 2023-11-01

**Soundness:** 3 good
**Presentation:** 3 good
**Contribution:** 3 good
**Rating:** 6
**Confidence:** 2

**Summary:**

This paper proposes a new algorithm to train a softmax tree of arbitrary structure to reduce the training and inference time for softmax layer. The tree structure is learned optimally by interleaving steps that grow the structure with steps that optimize the parameters of the current structure. The resulting softmax tree improves considerably the predictive accuracy while reducing the model size and inference time even further, as demonstrated in datasets with thousands of classes.

**Strengths:**

The paper provided detailed literature research and it also did detailed empirical comparison with baseline models.

The new model has much better training time and inference time.

**Weaknesses:**

In the result tables, number of parameters in these models are missing. So it's not clear if the new model has less parameters.

**Questions:**

Is GPU used in the model training and evaluation? How will the model might perform when using the GPU?

---

> ### Author Response · Authors · 2023-11-21
>
> W1.
>
> Since AST produces hard decisions in each decision node, an instance passes through a single root to leave path and gets classified in the leaf softmax. We provide the number of floating point operations to indicate that AST learns trees that classify samples (on average) faster and more efficiently.
>
> Q1.
>
> The current version of AST is implemented on the CPU (with parallelism).

---

### Official Review · Reviewer_u3Fd · 2023-11-07

**Soundness:** 2 fair
**Presentation:** 3 good
**Contribution:** 3 good
**Rating:** 5
**Confidence:** 4

**Summary:**

The paper presents a method of adaptively building tree structure for Softmax Tree. The method performs top-down tree growing: starting from a shallow tree, it grows the leaves with small subtrees in a BFS manner. After growing all leaves at one level, it performs joint optimization of the tree parameters. The method is compared with static tree structures for Softmax Trees, and with other tree-based methods for multiclass classification, over various datasets.

**Strengths:**

Determining an optimal tree structure for multi-class and multi-label tree-based methods is a challenging and important problem. The introduced method would only expand the leaf when the total loss doesn’t degrade (allowing a slight margin for degradation), which gives some confidence on the obtained tree. The experimental part shows that the learned tree structure improves over the static structure.

**Weaknesses:**

1. The experimental study is limited: the comparisons with other methods are provided only on a single Wiki-small dataset. From that, it’s not enough to judge on the comparison with other baselines.

2. The training time seems to be the main bottleneck of the method, its training is slower than for almost any other tree method (as reported in the paper). Probably because of that, applying the method on bigger datasets becomes infeasible. (Fair to say, that the same shortcoming applies for the original Softmax Tree, and the presented method seems to double the training time).

3. The method seems to be quite sensitive to hyperparameters, so in order to apply it method for a new problem, one has to perform some careful hyperparameter search to find a proper $\alpha$.

**Questions:**

1. LOMTree is mentioned in the paper as one of the baselines but I didn’t see any results for it.

2. Can the method produce degraded trees, when only a single tree path is getting expanded? Is there any guardrail for it?

3. Is the Wiki-small actually a multi-label dataset (the instances can have more than 1 label)? Was it somehow transformed to a multi-class problem?

---

> ### Author Response · Authors · 2023-11-21
>
> W1.
>
> At this work we show that learning structure of a tree using AST produces better results then initializing ST with complete tree randomly or using k-means. We provide a comparison on 4 datasets: Letter, ALOI, LSHTC1, Wiki-Small for classification and PTB dataset for language modeling on which AST consistently outperforms ST. Results of Zharmagambetov et al., 2021 shows that ST outperforms other baseline methods.
>
>
> W2.
>
> The goal of the proposed approach is to learn once an accurate model with very fast inference. The training time will be longer because of the more exploration and optimization steps but the significant gain in accuracy and inference time justifies the approach.
>
> To train AST efficiently we use warm starting in decision nodes and in leaf nodes. Furthermore, each expansion step and regular step requires up to 5 - 10 TAO iterations (depending on the dataset), while ST is typically trained for much longer. While training time is slower, it is comparable to other methods such as ST and MACH, and much faster compared to softmax or one-vs-all.
>
> W3.
>
> We provide an extensive study of hyperparameters of both ST and AST. For most datasets AST outperforms ST with default parameters ($\alpha = 0.5$, $\rho = 1.0$), while tuning $\alpha$ even further increase the gap between two methods. Values of $\lambda$ and $\mu$ can be tuned in the same way as for ST, by setting $\lambda = \mu$. What is more, good choice of $\alpha$ can compensate for not optimal choice of $\mu$.
>
>
> Q2.
>
> Yes, in principle it is possible for the algorithm to expand a tree only through one path, and thus produce such ``lopsided'' trees. But we think that there is nothing inherently wrong with such trees in that the underlying true distribution of points can favor such structure, if for example, a certain group of classes can be easily separated from the rest in each decision node. Having said that, we do not observe such behavior in our experiments, and since especially our random initialization (based on random direction and median split) has some bias towards balanced trees.
>
> Q3.
>
> We take this dataset from (Joshi et al. NIPS2017). The authors there convert this multi-label dataset to multi-class format by replicating the instances belonging to different class labels.

---

> > ### Comment · Reviewer_u3Fd · 2023-11-22
> >
> > Thanks for the reply; a couple of follow up questions:
> >
> > W1: Have you performed the comparison of AST vs ST on ODP dataset as in [Zharmagambetov et al., 2021]?
> >
> > W3: In the result tables I couldn't see any supports for the claim that the AST outperforms ST when \rho=1. The only such comparison I could find is in the Table 2, showing that AST is not better.

---

> > > ### Author Response · Authors · 2023-11-22
> > >
> > > Thank you for your response.
> > >
> > > W1: No, however we can add it to camera ready version of the paper.
> > >
> > > W3: Table 2 language modelling (right) all AST models are trained with $\rho=1.0$. For LSHTC1 and for Letter dataset it is also a case, however the gap between the results is not that large for $\rho = 1.0$, so we choose to report better models.

---

### Official Review · Reviewer_FA55 · 2023-11-08

**Soundness:** 2 fair
**Presentation:** 2 fair
**Contribution:** 2 fair
**Rating:** 3
**Confidence:** 3

**Summary:**

The paper introduces Adaptive Softmax Trees (ASTs), an extension of Softmax Trees. The original Softmax Tree is a mix of hard and soft decision tree algorithms. The internal nodes (here called decision nodes) contain hard routers and leaf nodes softmax estimators. Originally, Softmax Tree has a predefined tree structure and uses the Tree Alternating Optimization (TAO) algorithm for training its node parameters. The idea behind the proposed extension is to build the tree in an iterative manner, starting from the shallow predefined tree (trained with TAO) and in each step by trying to expand a leaf node into new subtrees, trained using the same TAO algorithm, that is added to the tree if it yields the improvement in the optimized objective. If the subtree is added, the new tree is again retrained using the TAO algorithm. The attractiveness of the proposed approach is evaluated on text classification and language modeling task and compared against a few baselines that also use linear classifiers. The results confirm the superiority of AST over the baselines and original ST in terms of predictive performance and inference times.

**Strengths:**

1. The empirical comparison seems to prove the attractiveness of the proposed approach.

**Weaknesses:**

1. The novelty of the paper is very limited as the proposed extension is quite simple and of a heuristic nature, as there is also no theoretical contribution accompanying it.

2. Section 4 copies a lot of text from the paper of Zharmagambetov et al., 2021, which itself is not such a big issue for me, but unfortunately, I find both explanations hard to follow and missing important details. The biggest issues, in my opinion, are:
    - $\mathbf{y}_n$ is not defined,
    - output of the $\tau(\mathbf{x}_n; \Theta)$ is also not clearly defined,
    - it's not clear what is a tree structure, I understand that it's given since it's parameterized by $\Delta$ and $k$, it seems that classes in the leaves can be redundant. I understood that the class assignment is either random or obtained by some k-means clustering.

3. It seems that while the ST provides very fast inference due to hard tree routing, it is costly to train, especially the proposed AST variant that repeats TAO training multiple for each and after each expansion of the tree.

4. > Training HSM-based language models is efficient (usually logarithmic in vocabulary size), but it leads to no speedup at inference time: during prediction, an input instance is propagated to all the leave

     This statement is not true, HSM structure allows efficient retrieval of top-k classes or all the classes about the given threshold of marginal probability $P(y | \boldsymbol{x})$ by applying a proper tree search algorithm.

5. The strength of softmax and hierarchical softmax is that they are fully differentiable and can be easily used as a part of more complex architectures. They also aim to provide calibrated estimates of conditional class probabilities $P(y | \boldsymbol{x})$. Hierarchical softmax also speeds up both training and inference. As in the case of ST/AST, performance/speed-up trade-off can be easily controlled by selecting the proper tree structure. The ST/AST, while providing superior predictive performance, seem not to allow end-to-end training and do not aim to provide accurate probability estimates, which severely limits their applications. I belive the relevance of this work.

6. I got the impression that the AST required a lot of hyperparameter tuning before it achieved better results than ST.

NITs:
1. There are some related works that authors might consider discussing:
    - A quite recent algorithm that also mixes hard trees and soft trees (could serve as another baseline): *Sun, W., Beygelzimer, A., Iii, H. D., Langford, J., and Mineiro, P. (2019). Contextual memory trees.*
    - Variant of HSM that builds tree structure online: *Beygelzimer, A., Langford, J., Lifshits, Y., Sorkin, G. B., and Strehl, A. L. (2009). Conditional probability tree estimation analysis and algorithms*
    - Generalization of HSM from multi-class to the multi-label case: *Wydmuch, M., Jasinska, K., Kuznetsov, M., Busa-Fekete, R., and Dembczynski, K. (2018). A no-regret generalization of hierarchical softmax to extreme multi-label classification*

2. Why not include hierarchical softmax in the empirical comparison for text classification? What variant is used on PTB task? Many variants are possible, e.g., the most computationally performant with a hamming tree, or popular in neural networks, two-level hierarchical softmax, which provides less speed-up in terms of complexity but is GPU-friendly and usually very close to flat softmax in terms of predictive performance.
3. No citation for Penn Treebank (PTB) dataset

**Questions:**

I would be happy to see the authors respond to my critique from the weaknesses section.

Additional questions:
- How the structure of the new subtree is decided?
- Is reported training time, a clock time, or CPU time (does it take into parallelism account)?

---

> ### Author Response · Authors · 2023-11-21
>
> W1.
>
> The proposed extension fills an important gap in the original ST model. Since the original ST assumes an initial tree structure of a given depth, it cannot explore deeper tree structures. It is also sensitive to initial model parameters, necessitating the use of various heuristic-based initialization. Our proposed approach effectively addresses these two shortcomings by cleverly expanding the tree where needed and globally reoptimizing the whole model. Each step of our algorithm has a guarantee of monotonically decreasing the objective function value.
>
> W2.
>
> Because the algorithm is not widely popular we choose to review the algorithm in more detail. Therefore, our explanation is similar (but not a copy) to the one in (Zharmagambetov et al., EMNLP 2021)
>
> In Section 4, we define a dataset $\{(x_n, y_n)\}_{n=1}^N \subset R^D \times \{1,...,K\}$, where $y_n$ is ground truth target label.
>
> The output of a Softmax tree $\tau(x; \theta)$ is a predicted label for an instance $x$.
>
> To obtain initial Softmax Trees in our algorithm we use random initialization, without doing any heuristic-based $k$-means unlike in (Zharmagambetov et al, 2021).
> The number of classes in each leaf softmax is at most $k$. If the set of training points reaching a given leaf has smaller number of classes than $k$, then the corresponding leaf softmax adjusts to this.
>
>
> W3.
>
> The goal of the proposed approach is to learn once an accurate model with very fast inference. The training time will be longer because of the more exploration and optimization steps but the significant gain in accuracy and inference time justifies the approach.
>
> To train an AST efficiently we use warm starting in decision nodes and in leaf nodes. Furthermore, each expansion step and regular step requires up to 5 - 10 TAO iterations (depending on the dataset), while the original ST is typically trained for much longer. While training time is slower, it is comparable to other methods such as ST and MACH, and much faster compared to softmax or one-vs-all.
>
> W4.
>
> While it is possible to prune the search space in HSMs when finding for a maximum probability leaf, many nodes/leaves has still be explored. This is unlike in our approach where one and only one leaf is reached during inference.
>
> W5.
>
> Yes, it is true that an HSM has the advantage of being differentiable and the support for end-to-end learning. But we do not propose ASTs as a drop-in replacement for HSMs. We view ASTs as standalone models for many-class classification problems with good enough accuracy and very fast inference. ASTs can also be trained on neural network (NN) features, and so be used as a last classifier layer in NNs (as we do in our language modeling experiments). And ASTs output correctly the probabilities, in that it is nonzero for a small subset of classes (for the classes in the reached leaf) and exactly zero for the rest.
>
> W6.
>
> For most datasets AST outperforms ST with default parameters ($\alpha = 0.5$, $\rho = 1.0$), while tuning $\alpha$ even further increase the gap between two methods. Values of $\lambda$ and $\mu$ can be tuned in the same way as for ST, by setting $\lambda = \mu$. What is more, good choice of $\alpha$ can compensate for not optimal choice of $\mu$.
>
>
> Q1.
>
> To expand we use a subtree of depth 1. We mention that depth 2 can be used to grow tree faster, however training each subtree requires more iterations.
>
> Q2.
>
> We report clock time taking into account parallelism.

---

> > ### Comment · Reviewer_FA55 · 2023-11-23
> > **Re: Official Comment by Authors**
> >
> > Thank you for your comments, which clarified for me some parts of the submission; however, I still have doubts about a few aspects:
> >
> > > The output of a Softmax tree $\tau(x; \theta)$ is a predicted label for an instance $x$.
> >
> > So in Equation (1) and (3), **bold** $\boldsymbol{y}_n$ should be normal $y_n$, or is there some reason it's bolded there?
> >
> > > And ASTs output correctly the probabilities, in that it is nonzero for a small subset of classes (for the classes in the reached leaf) and exactly zero for the rest.
> >
> > If I understand the ST algorithm correctly, the softmax in a single leaf is trained using only subsets of examples. So it seems to me that while indeed the value is some kind of probability, it is a biased estimate of $P(y | \boldsymbol{x})$.
> >
> > > We report clock time taking into account parallelism.
> >
> > So, is the training time reported in Table 2 a clock time when using multiple cores/threads? If yes, then I'm missing detailed information about the experimental setup that ensures that all the methods can use the same level of parallelism for fair comparison. If this is not possible, I would recommend reporting CPU time instead. I'm also missing the information about the hardware that was used to run these experiments.
> >
> > ---
> >
> > Overall, the authors' response does not change my opinion that their work is a simple algorithmic extension of ST, which itself has narrow applications, and because of that, both the contribution and relevance of this work are limited. Additionally, I believe the presentation of the paper could be improved. I keep my score unchanged.

---

### Comment · Area_Chair_DowD · 2023-11-22
**Author-Reviewer Discussion ends soon**

Dear Reviewers and Authors,

The discussion phase ends soon. Please check all the comments, questions, and responses and react appropriately.

Thank you!

Best, AC for Paper #1530